# Rhodomyrtone as a New Natural Antibiotic Isolated from *Rhodomyrtus tomentosa* Leaf Extract: A Clinical Application in the Management of Acne Vulgaris

**DOI:** 10.3390/antibiotics10020108

**Published:** 2021-01-22

**Authors:** Suttiwan Wunnoo, Siwaporn Bilhman, Thanaporn Amnuaikit, Julalak C. Ontong, Sudarshan Singh, Sauvarat Auepemkiate, Supayang P. Voravuthikunchai

**Affiliations:** 1Division of Biological Science, Faculty of Science and Natural Product Research Center of Excellence, Prince of Songkla University, Hat Yai, Songkhla 90112, Thailand; dppoom.girl@gmail.com (S.W.); siwaporn8195@gmail.com (S.B.); jula_601_f@hotmail.com (J.C.O.); sudarshansingh83@hotmail.com (S.S.); 2Department of Pharmaceutical Technology, Faculty of Pharmaceutical Sciences, Prince of Songkla University, Hat Yai, Songkhla 90112, Thailand; chomchan.a@psu.ac.th; 3Department of Pathology, Faculty of Medicine, Prince of Songkla University, Hat Yai, Songkhla 90112, Thailand; Sauvarata@gmail.com

**Keywords:** antibiotic, antimicrobial, rhodomyrtone, liposome, natural products, clinical trial

## Abstract

Rhodomyrtone, a plant-derived principal compound isolated from *Rhodomyrtus tomentosa* (Myrtaceae) leaf extract, was assessed as a potential natural alternative for the treatment of acne vulgaris. The clinical efficacy of a 1% liposomal encapsulated rhodomyrtone serum was compared with a marketed 1% clindamycin gel. In a randomized and double-blind controlled clinical trial, 60 volunteers with mild to moderate acne severity were assigned to two groups: rhodomyrtone serum and clindamycin gel. The volunteers were instructed to apply the samples to acne lesions on their faces twice daily. A significant reduction in the total numbers of acne lesions was demonstrated in both treatment groups between week 2 and 8 (*p* < 0.05). Significant differences in acne numbers compared with the baseline were evidenced at week 2 onwards (*p* < 0.05). At the end of the clinical trial, the total inflamed acne counts in the 1% rhodomyrtone serum group were significantly reduced by 36.36%, comparable to 34.70% in the clindamycin-treated group (*p* < 0.05). Furthermore, a commercial prototype was developed, and a clinical assessment of 45 volunteers was performed. After application of the commercial prototype for 1 week, 68.89% and 28.89% of volunteers demonstrated complete and improved inflammatory acne, respectively. All of the subjects presented no signs of irritation or side effects during the treatment. Most of the volunteers (71.11%) indicated that they were very satisfied. Rhodomyrtone serum was demonstrated to be effective and safe for the treatment of inflammatory acne lesions.

## 1. Introduction

The emergence and spread of antibiotic resistance among clinically relevant pathogens demand alternative and effective novel antibacterial drugs to address the current antibiotic resistance crisis. Over the past decade, consumers worldwide have become more aware of the use of natural products for the treatment and prevention of infectious diseases due to their efficacy and safety. Natural products are a rich source of antibacterial compounds. A drug derived from an herbal source can play a significant role in the future healthcare system. *Rhodomyrtus tomentosa* (Aiton) Hassk., an evergreen shrub belonging to the family Myrtaceae, has been used in traditional medicine in Southeast Asian countries. Rhodomyrtone, an acylphloroglucinol compound isolated from *R. tomentosa*, has excellent bioactive properties against a wide range of Gram-positive bacteria [1,2], with a novel mechanism of action [3]. Antibiotics with novel mechanisms of action minimize the probability of pre-existing resistance and also prevent the rapid development of resistance among pathogens [4]. Emerging evidence suggests that rhodomyrtone from *R. tomentosa* might represent a possible natural alternative for the treatment of bacterial infections [5,6,7].

Acne vulgaris is a common skin problem characterized by the abnormality of the pilosebaceous unit in areas such as the face, neck, and upper trunk. Moreover, severe cases of acne might persist for many years in teenagers and adults. The attachment of anaerobic bacteria *Propionibacterium acnes* to hair follicles, followed by the colonization of follicular ducts, is the mechanism of acne vulgaris pathogenesis [8,9]. Topical anti-acne preparations act by reducing the follicular population of *P. acnes* and influencing the ability to generate proinflammatory molecules [10]. Several over-the-counter acne products, including gels, cleansing lotions, foams, and treatment kits, are available for the treatment of mild to moderate acne. Medical treatment of mild acne requires topical therapies, including benzoyl peroxide, retinoic acid/azelaic acid, erythromycin, and clindamycin [11]. Moderate acne responds well to combination therapy between topical and oral antibiotics, along with oral contraceptive pills [12,13]. Current acne medication may cause mild to severe adverse side effects from chemical agents, for example, skin irritation from prolonged usage of benzoyl peroxide or retinoic acid, including erythema and peeling of facial skin. Moreover, treatment may encounter problems from biofilm-forming or antibiotic-resistant bacteria [14]. Therefore, anti-acne therapy requires the development of a safe and clinically effective natural antibiotic.

Medicinal plants have a long history of being well-known for biological properties, including antimicrobial activity. In recent decades, certain extracts of plants have been targeted as alternative therapeutic agents for acne treatment [15]. For example, pharmaceutical formulations with plant-derived antimicrobials, including *Garcinia mangostana,* [16,17] and *Camellia sinensis* [18], have been used as topical medicine with wide acceptance as cosmetic products. In addition, a number of purified natural compounds [19] and essential oils [15] derived from plants have been developed as anti-acne agents. Moreover, anthraquinones from *Rubia cordifolia* [20] and *Melaleuca alternifolia* oil nanoparticles [21,22] have also demonstrated efficacy in managing acne. To maximize healthcare resources for acne treatment, rhodomyrtone, an advanced natural acne-fighting principal compound from *Rhodomyrtus tomentosa* leaves, has been extensively studied. The minimal inhibitory concentration (MIC) that inhibits 90% growth of *P. acnes*, an etiologic agent of acnes, was 0.5 µg/mL [5]. In addition, our previous work demonstrated that the compound significantly decreased *P. acnes* biofilm formation and inhibited bacterial viability within the mature biofilm as well as significantly reduced the production of lipase [7]. Minimal inhibitory concentrations for multidrug-resistant *Staphylococcus aureus* and *Staphylococcus epidermidis*, associated microorganisms in acne-affected areas, ranged from 1–2 µg/mL, while the minimal bactericidal concentrations (MBCs) were 4–8 µg/mL. Moreover, recent toxicological reports on invertebrate and vertebrate models suggested the safety of rhodomyrtone [23].

The present study was designed to examine clinical evidence on the efficacy and safety of 1% liposomal encapsulated rhodomyrtone serum formulation in acne treatment. A randomized, double-blind clinical trial was performed to compare the efficacy of the formulation with 1% clindamycin gel, a commonly prescribed antibiotic for topical acne treatment. In addition, we developed a commercial prototype for the treatment of inflammatory acne lesions. The clinical study and subject satisfaction with the prototype were evaluated.

## 2. Results and Discussion

Information on volunteer characteristics related to acne is presented in Table 1. Thirty volunteers initially enrolled in each treatment group: 1% liposomal encapsulated rhodomyrtone serum group and 1% clindamycin gel-treated group completed the whole study protocol (Figure 1). Most of the volunteers were students at Prince of Songkla University with an average age of 21 years. Initially, the baseline general characteristics of all volunteers were not significantly different between the treatment groups (*p* > 0.05). In addition, no differences in the number of acne lesions or the Investigator Global Assessment (IGA) scale between the groups were observed (*p* > 0.05) (Table 2).

All subjects showed no signs of irritation or side effects in response to the tested topical agents. Safety evaluation of all volunteers did not reveal severe cases of skin allergy during the treatment. However, some subjects experienced itching around the acne lesions during the early stage of the treatment. The severity of these events earned a mild score in all groups: 3.33% in 1% liposomal encapsulated rhodomyrtone serum, and 6.67% in 1% clindamycin gel (data not presented). Most of the volunteers’ responses (50.00% and 43.30%) indicated that they were moderately satisfied (score = 2) and 36.70% and 40.00% were very satisfied (score = 3) with the 1% liposomal encapsulated rhodomyrtone serum and 1% clindamycin gel, respectively (Table 3).

Reduction in acne lesions after treatment for 8 weeks is presented in Figure 2. All treatment agents resulted in a similar reduction in the numbers of acne lesions. Significant reductions in the total numbers of acne lesions were demonstrated in all treatment groups, compared with the baseline, from week 2 to week 8 (Figure 2A) (*p* < 0.05). For non-inflammatory lesions (Figure 2B), a reduction in total acne counts was observed in all treatment groups. It was noted that volunteers with 1% liposomal encapsulated rhodomyrtone serum demonstrated improvement in inflammatory lesions relative to the baseline within a few days. We have previously reported that 1% liposomal encapsulated rhodomyrtone serum had better efficacy than 1% clindamycin, 20% azelaic acid, and 2.5% benzoyl peroxide when employed in acne treatment [10]. At week 8, total inflamed acne lesion counts in the 1% liposomal encapsulated rhodomyrtone serum-treated group were significantly reduced by 36.36%, compared with 34.70% in the 1% clindamycin gel group (*p* < 0.05) (Figure 2C). A volunteer case with mild-severity acne vulgaris demonstrated a remarkable reduction in the numbers of acne lesions, with acne lesions almost eradicated at the end of the treatment (Figure 3). Liposomal encapsulated rhodomyrtone serum treatment in a volunteer with moderate-severity acne vulgaris resulted in fewer numbers of acne lesions. In addition, a change in severity towards mild severity was observed at the end of treatment, as represented in Figure 4. No significant differences in the IGA scale change among the two treatment groups were observed (*p* > 0.05) (Table 4).

In addition to its anti-acne properties, rhodomyrtone presented extremely potent and broad Gram-positive antibacterial activity, with an MIC and MBC at 0.39 to 0.78 µg/mL and 0.39 to 12.5 µg/mL, respectively, which are comparable to last-resort antibiotics in glycopeptide and lipopeptide groups [1]. Both its transient binding mode to phospholipid head groups, leading to distortion of lipid packing, membrane fluidization, and induction of membrane curvature, and its ability to form protein-trapping membrane vesicles are unique, making it an attractive new antibiotic candidate with a novel mechanism of action [3]. Rhodomyrtone was also demonstrated to modulate the transcription of genes involved in the diaminopimelate biosynthetic pathway, associated with the biogenesis of the cell envelope, transporter proteins, and nucleotide metabolism [24]. The compound exhibits activity against multidrug-resistant organisms and capsule forming- and endospore-producing bacteria [25,26]. In addition, the ability to hinder bacterial adhesion to human HaCaT keratinocytes [27], inhibit biofilm production, and kill pathogens within a biofilm was demonstrated [7,28].

*Propionibacterium acnes* induces inflammation through Toll-like receptor 2 (TLR2) activation on keratinocytes, leading to the release of pro-inflammatory cytokines such as TNF-α, IL-6, and IL-8 [29]. Following the treatment, inflamed acne lesions distinctly improved. Currently, many cytokines, chemokines, and other pro-inflammatory mediators are gaining a great deal of attention by many scientists seeking to reveal breakthroughs in the treatment of acne [30]. Evidence suggests that rhodomyrtone significantly decreases inflammatory gene expression and the expression and secretion of inflammatory proteins by modulating MAP kinase and NF-κB signaling pathways [31]. The bioactive compound could enhance the expression of pro-inflammatory molecules, including IL-6, and iNOS in simulated THP-1 monocytes with heat-killed methicillin-resistant *S. aureus* [32]. In addition, the efficiency of a phospholipid, a natural ingredient in liposomes, has been previously shown to maintain moisturize effect on the skin [33] as well as act as an anti-inflammatory agent [34]. The liposome formulation contains some fatty acids that provide skin hydration, resulting in enhanced efficacy of acne treatment in volunteers. Liposomal encapsulated rhodomyrtone serum demonstrated better results than clindamycin gel. Furthermore, volunteers’ responses suggest the potential of the prototype for commercialization for the treatment of inflammatory acne vulgaris.

A clinical trial on a commercial prototype was assessed in 45 subjects for a period of 1 week. Volunteer baseline characteristics related to acne are displayed in Table 5. All volunteers presented no signs of irritation or side effects during the treatment. The clinical evaluation of the prototype demonstrated 68.89% complete reduction and 28.89% improvement in inflammatory acne lesions, with only one non-improved case (Table 6 and Appendix A). In most cases, the improvement in inflamed acne became obvious within 3 days following the treatment, with no severe irritation observed (Figure 5). Most subjects had IGA scale improvements, 40.00% for a 2-score improvement and 42.22% for a 1-score improvement (Table 6 and Appendix A). Satisfaction level assessment demonstrated that 71.11% and 28.89% were very satisfied and moderately satisfied (Table 3).

## 3. Materials and Methods

### 3.1. Materials

L-phosphatidyl choline (Sigma, St. Louis, MO, USA), cholesterol (Sigma, Fluka, Tokyo, Japan), ethanol (Merck, Darmstadt, Germany), DOW Corning^®^ RM 2051 and DOW Corning^®^ 9045 silicone (Dow Corning Corporation, Midland, MI, USA), methylparaben, and propylparaben (P.C. Center Co. Ltd., Bangkok, Thailand) were used. The leaves of *Rhodomyrtus tomentosa* were collected from Bermang locality, Raman District, Yala Province, Southern Thailand. The classified reference voucher specimen of *R. tomentosa* (NPRC0057) was deposited at Faculty of Traditional Thai Medicine, Prince of Songkla University, Hat Yai, Songkhla, Thailand. Rhodomyrtone, a pure compound from the leaves of *R. tomentosa* (family *Myrtaceae*), was isolated according to our previous report with slight modification [35,36]. Briefly, the dried leaf powder of *R. tomentosa* was extracted twice with 95% ethanol at room temperature for 5 days. The extract was evaporated by using a rotary evaporator (BUCHI Rotavapor R-114, Flawil, Switzerland). The dried extract was subjected to column chromatography to elute rhodomyrtone (Figure 6).

### 3.2. Liposomal Encapsulated Rhodomyrtone Serum

Liposomal encapsulated rhodomyrtone was formulated as per our previous report [6]. Briefly, liposomal encapsulated rhodomyrtone was prepared by a modified ethanol injection method. Rhodomyrtone was dissolved in absolute ethanol to an obtained concentration of 100 mg/mL. The lipid phase was prepared with 60 μmol/mL of total lipid concentration by dissolving phosphatidylcholine and cholesterol in the ratio of 4:1 (*w*/*w*) in 10 mL of ethanol. Ten microliters of rhodomyrtone in ethanol was dissolved in the lipid phase, and 10 mL of Milli-Q water was warmed in a water bath separately until the temperature of both phases reached 60 °C, followed by addition of both and sonication for 30 min. Liposomal encapsulated rhodomyrtone serum was prepared in a serum base (DOW Corning^®^ RM 2051, DOW Corning^®^ 9045 silicone, preservatives and water) to form the final concentration of 1% *w*/*w* of rhodomyrtone liposomal encapsulated serum.

### 3.3. Clindamycin Gel^®^

A 1% clindamycin gel^®^ (Clindalin gel, Clinda, Bangkok, Thailand), each gram of gel containing 10 mg of clindamycin phosphate, was obtained from the Songklanagarind Hospital, Thailand.

### 3.4. Clinical Study Design

The efficacy and safety of 1% liposomal encapsulated rhodomyrtone serum formulation and 1% clindamycin gel were comparatively studied in a randomized, double-blind clinical controlled trial conducted at Natural Product Research Center of Excellence, Faculty of Science, Prince of Songkla University, Thailand. Sixty volunteers were randomly given treatment with liposomal encapsulated rhodomyrtone serum and clindamycin gel for 8 weeks (Figure 1). All formulations were concealed until the end of the study period, and the code numbers were later identified by a blind third-party evaluator.

#### 3.4.1. Subject Selection

A total of 60 volunteers with mild to moderate acne vulgaris including 30 males and 30 females, aged between 18 to 25 years, were enrolled. Volunteers were well-versed to avoid using other anti-acne products in the interim. Exclusion criteria were set for volunteers with the following parameters: other facial skin diseases or dermatological conditions that required the use of topical or systemic therapy; redness, dryness, itching, and skin peeling on the face; known congenital diseases, chronic diseases, malnutrition, pregnancy, or hyperandrogenism; severe inflamed acne in more than five spots; acne history resulting from facial products; undergoing treatment with topical or systemic antibiotics, and retinoid or topical steroids on the face within the past 4 weeks; consuming hormones such as progesterone or estrogen within the past 3 months; and sensitivity to clindamycin or other excipients in the study medication.

Before starting the treatment, all volunteers signed an informed written consent statement. Prior to the clinical trial, all volunteers were tested for allergy by applying an assigned agent on the inner side of the arm. Volunteers with an allergic reaction were advised to withdraw from the study. The volunteers were instructed to apply tested agents to acne lesions on their face twice daily (in the morning and evening) for a period of up to 8 weeks.

#### 3.4.2. Efficacy Parameters

Efficacy evaluation of the treatment was monitored during weeks 2, 4, and 8. Clinical assessment was based on a lesion count and an acne grading system as described by the IGA [37]. A skin doctor assessed the volunteers once a day by examining and feeling the skin. Total acne lesion areas were photographed to ensure the coherence of each evaluation. Acne lesions were counted and categorized into different types of non-inflammatory (open comedones, closed comedones) and inflammatory (papule, pustule, nodule) conditions. Total IGA scale was assessed and acne grading was assigned to each patient at every appointment as follows: 0 equals clear skin, no inflammatory or non-inflammatory lesions; 1 equals almost clear skin, rare non-inflammatory lesions with no more than one small inflammatory lesion; 2 equals mild severity, greater than grade 1, some non-inflammatory lesions with no more than a few inflammatory lesions (papules, pustules only, no nodular lesions); 3 equals moderate severity, greater than grade 2, up to many non-inflammatory lesions and may have some inflammatory lesions, but no more than one small nodular lesion; and 4 equals severe, greater than grade 3, up to many non-inflammatory and inflammatory lesions but no more than a few nodular lesions. To ensure accuracy, the volunteer assessment and case management were conducted by the same evaluator throughout the treatment period. Efficiency evaluation was carried out by calculating the percentage reduction of non-inflammatory and inflammatory acne lesions.

#### 3.4.3. Safety Evaluation

Safety evaluation was performed at each clinical visit with the following parameters: 0 equals absent; 1 equals mild; 2 equals moderate; and 3 equals severe and skin allergy (erythema, dryness, burning, and pruritus). Patients’ Global Assessment was evaluated based on their response to acne treatment with the agents. Satisfaction was recorded according to the following scale: 0 equals dissatisfaction; 1 equals slightly satisfied; 2 equals moderately satisfied; and 3 equals very satisfied.

#### 3.4.4. Ethical Considerations

The treatment protocol was carried out by following the guidance of the ethical principles from the Declaration of Helsinki and good clinical practices and in compliance with international regulatory requirements. The patients’ personal information was anonymized so that the actual individuals could not be identified. The study protocol was approved by Songklanagarind Hospital Review Committee (Project Number 56-329-05-1) as per the international guideline for a clinical study.

### 3.5. Customer Evaluation

A total of 45 healthy volunteers with almost clear to mild acne vulgaris were informed about the study details and provided written consent. All subjects were recommended not to take any oral anti-acne drugs during the study, and allergy testing was performed on the inner side of the arm. Clinical evidence on the efficacy and safety of the treatment was monitored for 7 days on the basis of clinical assessment as outlined in Section 3.4.2 and Section 3.4.3. Prior to and at the end of the study, the volunteers were assessed for inflammatory acne lesions and IGA scale. In addition, volunteer satisfaction was evaluated after using the commercial prototype.

### 3.6. Statistical Analysis

The variable characteristics between each group and the number of acne lesions at the baseline to the end of the treatment were evaluated using one-way ANOVA. A within-group comparison of changes in non-inflammatory and inflammatory acne lesions was estimated by paired t-test. Significant differences were assessed with p-values less than 0.05. A between-group comparison of the efficacy was assessed using analysis of covariance (one-way ANOVA using SPSS statistical software), with the combined numbers of acne lesions at the baseline as a covariate.

## 4. Conclusions

Rhodomyrtone product has been demonstrated to be an effective nature-derived antibiotic for treating acne lesions, especially inflammatory lesions, with comparable efficacy to clindamycin gel. Rhodomyrtone serum has been demonstrated to be effective and safe for the treatment of inflammatory acne lesions. In addition to infection control, use in human volunteers clinically demonstrated good healing activity and skin clarification. All subjects showed no signs of irritation or long-term undesirable side effects. We believe that a rhodomyrtone-containing product could be used as an alternative agent for acne treatment.

## Figures and Tables

**Figure 1 antibiotics-10-00108-f001:**
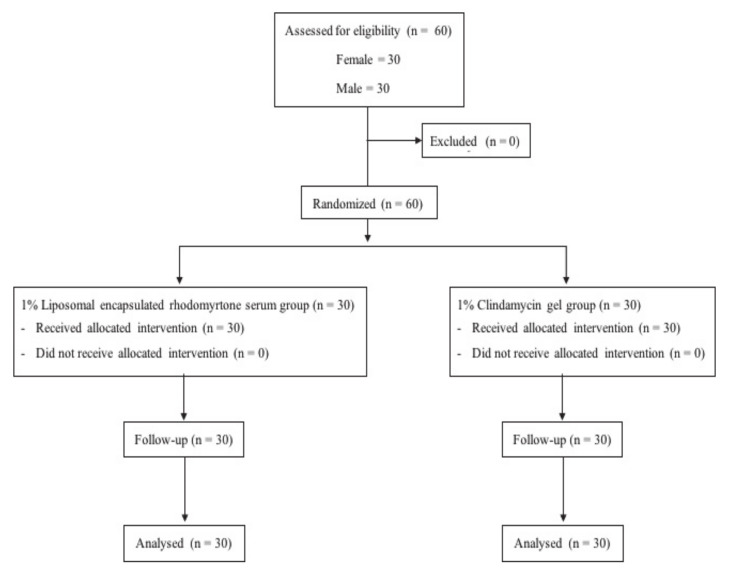
Clinical study design.

**Figure 2 antibiotics-10-00108-f002:**
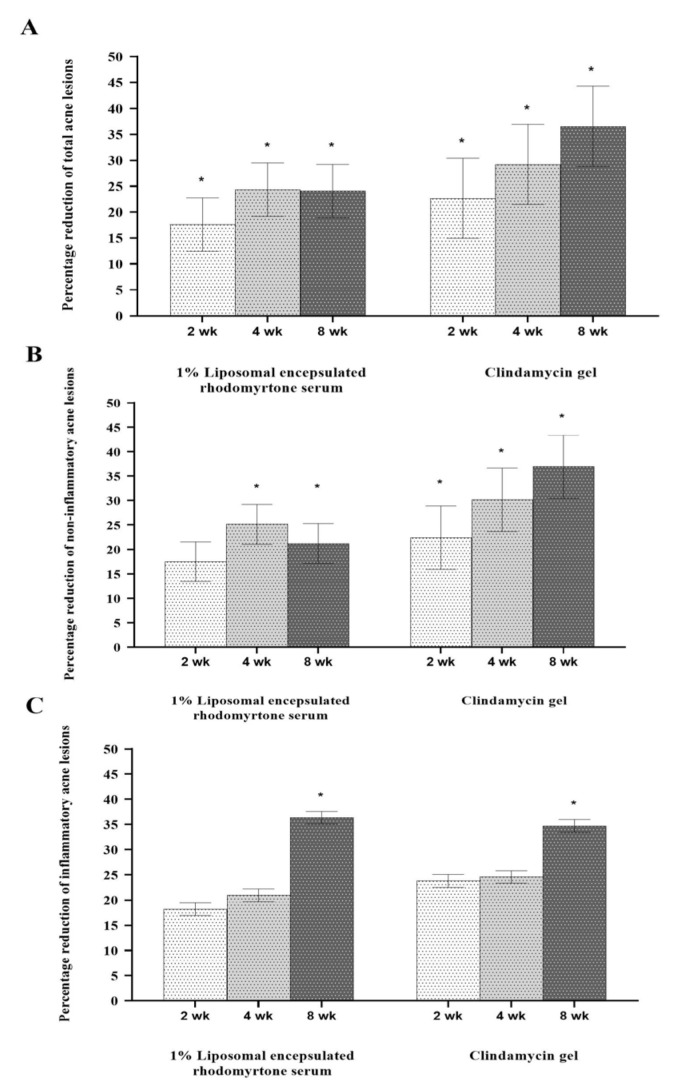
Percentage reduction in mean number of acne lesions after treatment with 1% liposomal encapsulated rhodomyrtone serum and 1% clindamycin gel at weeks 2, 4, and 8. Total acne lesions (**A**), non-inflammatory acne lesions (**B)**, and inflammatory acne lesions (**C)**. * Statistically different between the baseline and each time point of treatments (*p* < 0.05).

**Figure 3 antibiotics-10-00108-f003:**
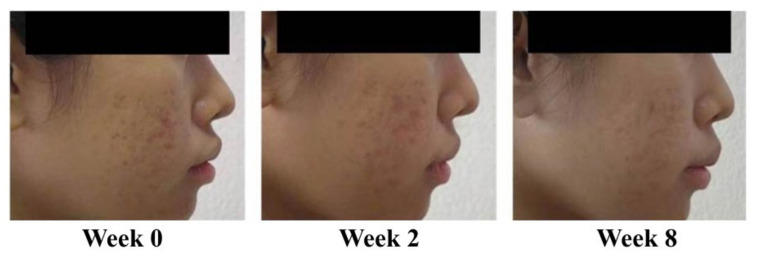
Case study: a 23-year-old female with mild-severity acne vulgaris at the baseline (IGA (Investigator Global Assessment) scale = 2, some non-inflammatory lesions with no more than a few inflammatory lesions). By week 8, following the treatment with 1% liposomal encapsulated rhodomyrtone serum twice daily, a change in severity to almost acne vulgaris-free occurred (IGA scale = 1, rare non-inflammatory with no more than one small inflammatory lesion).

**Figure 4 antibiotics-10-00108-f004:**
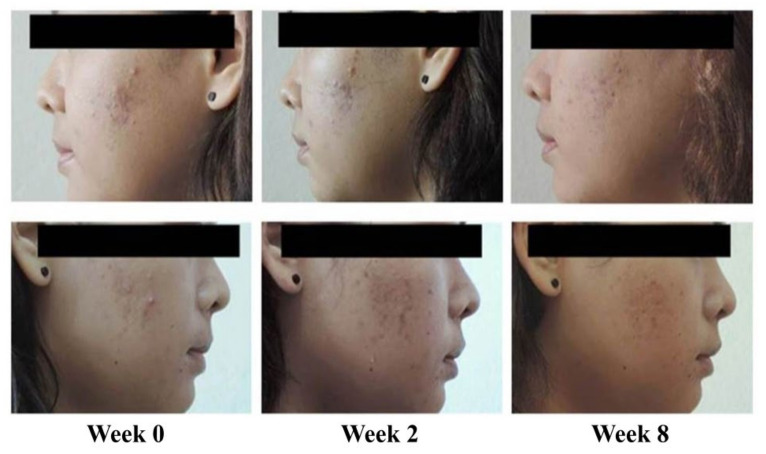
Case study: a 21-year-old female with moderate-severity acne vulgaris at the baseline (IGA scale = 3, up to many non-inflammatory lesions and may have some inflammatory lesions). By week 8, following the treatment with 1% liposomal encapsulated rhodomyrtone serum twice daily, a change in severity of acne vulgaris to mild severity occurred (IGA scale = 2, some non-inflammatory lesions with no more than a few inflammatory lesion).

**Figure 5 antibiotics-10-00108-f005:**
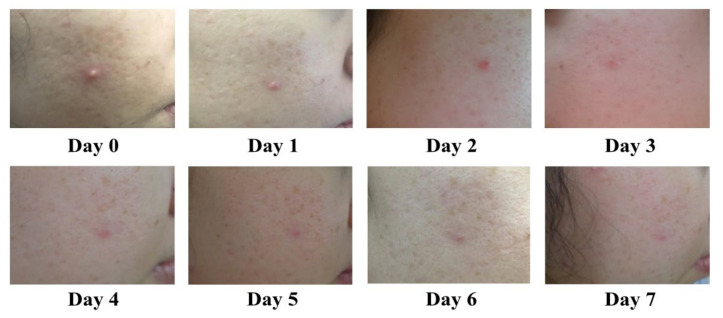
Representative photographs demonstrate improvement of acne in a volunteer treated with the commercial prototype, twice daily for 7 days.

**Figure 6 antibiotics-10-00108-f006:**
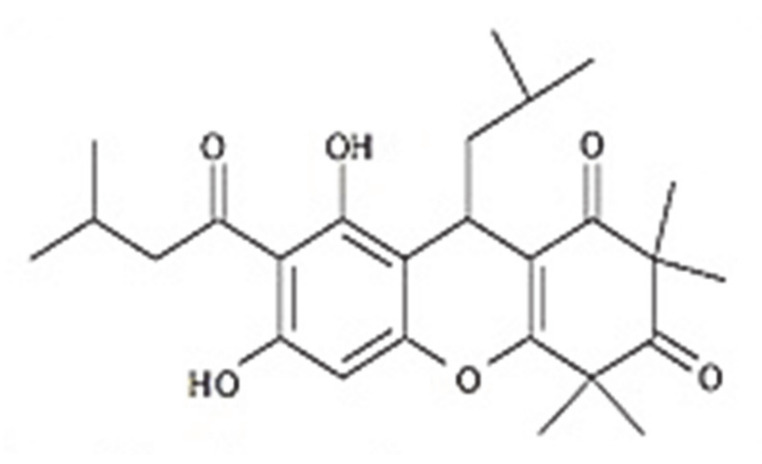
Chemical structure of rhodomyrtone.

**Table 1 antibiotics-10-00108-t001:** Baseline volunteer characteristics.

Clinical Parameter	Total (*n* = 60)	1% Liposomal Encapsulated Rhodomyrtone Serum (*n* = 30)	1% Clindamycin Gel (*n* = 30)	*p*-Values
Age (year)	21.23 ± 2.08 ^a^	21.10 ± 2.02	21.37 ± 2.17	0.625
Age starts having acne (year)	15.38 ± 2.75	15.03 ± 1.90	15.73 ± 3.40	0.331
Family with history of acne	46.00 (76.67) ^b^	23.00 (76.66)	23.00 (76.66)	1.000
**Patient History of Acne Treatment**				
Used to	38.00 (63.33)	19.00 (63.33)	19.00 (63.33)	1.000
See doctor for acne treatment	21.00 (35.00)	11.00 (36.67)	10.00 (33.33)	0.787
Use commercial drug	27.00 (45.00)	14.00 (46.67)	13.00 (43.33)	0.790
Use both	10.00 (16.67)	6.00 (20.00)	4.00 (13.33)	0.488
**Use Facial Cleanser**				
Ordinary facial cleanser	49.00 (81.67)	23.00 (76.67)	26.00 (86.67)	0.317
Acne facial cleanser	3.00 (5.00)	1.00 (3.33)	2.00 (6.67)	1.000
Soap	8.00 (13.33)	6.00 (20.00)	2.00 (6.67)	0.250
**Use Other Product**				
Facial skin care	38.00 (63.33)	20.00 (66.67)	18.00 (60.00)	0.592
Sunscreen	37.00 (61.02)	19.00 (63.33)	18.00 (60.00)	0.791
Facial skin whitening	1.00 (1.67)	1.00 (3.33)	0	1.000

^a^ Values in the same row are means ± SD in each group; ^b^ Numbers of patients (percentage).

**Table 2 antibiotics-10-00108-t002:** Quantity of acne lesions and Investigation’s Global Assessment (IGA).

Number of Acne Lesions at Beginning	Total (*n* = 60)	1% Liposomal Encapsulated Rhodomyrtone Serum (*n* = 30)	1% Clindamycin Gel (*n* = 30)	*p*-Values
Total acne lesions	47.02 ± 30.12 ^a^	45.03 ± 33.77	49.00 ± 26.40	0.614
Non-inflammatory acne lesions	38.53 ± 26.45	36.60 ± 29.85	40.47 ± 22.91	0.576
Open comedones	4.15 ± 6.26	4.03 ± 6.76	4.27 ± 5.83	0.887
Closed comedones	34.38 ± 23.20	32.57 ± 25.48	36.20 ± 20.95	0.549
Inflammatory acne lesions	8.48 ± 7.77	8.43 ± 8.48	8.53 ± 7.14	0.961
Papule	7.25 ± 6.61	6.67 ± 6.26	7.83 ± 7.01	0.499
Pustule	1.22 ± 2.54	1.73 ± 3.30	0.70 ± 1.29	0.119
Nodule	0.02 ± 0.13	0.03 ± 0.18	0	0.326
**Investigator Global Assessment (IGA)**				
1	5.00 (8.33) ^b^	1.00 (3.33)	4.00 (3.33)	
2	30.00 (50.00)	18.00 (60.00)	12.00 (40.00)	
3	25.00 (41.67)	11.00 (36.67)	14.00 (36.67)	
4	0	0	0	
IGA scale	2.33 ± 0.62	2.33 ± 0.55	2.33 ± 0.71	0.186

^a^ Values in the same row are means ± SD in each group; ^b^ Numbers of patients (percentage).

**Table 3 antibiotics-10-00108-t003:** Percentage of volunteers’ responses to treatment.

Satisfaction Level	1% Liposome Encapsulated Rhodomyrtone Serum	1% Clindamycin Gel	Commercial Prototype
Very satisfied	36.70 ^a^	40.00 ^a^	71.11 ^b^
Moderately satisfied	50.00 ^a^	43.30 ^a^	28.89 ^b^
Slightly satisfied	13.30 ^a^	16.70 ^a^	0

^a^ Number of volunteers = 30; ^b^ Number of volunteers = 45.

**Table 4 antibiotics-10-00108-t004:** Investigator’s Global Assessment (IGA) scale.

Samples	IGA Scale	Numbers of Volunteers (%)		
		Week 2	Week 4	Week 8
**1% Liposomal encapsulated** **rhodomyrtone serum**	deteriorated (2 score)	0	0	0
	deteriorated (1 score)	3 (10.00) ^a^	3 (10.00)	2 (6.67)
	no change	19 (63.33)	23 (76.67)	21 (70.00)
	improved (1 score)	8 (26.67)	3 (10.00)	6 (20.00)
	improved (2 score)	0	1 (3.33)	1 (3.33)
**1% Clindamycin gel**	deteriorated (2 score)	0	0	0
	deteriorated (1 score)	3 (10.00)	4 (13.33)	4 (13.33)
	no change	22 (73.33)	18 (60.00)	15 (50.00)
	improved (1 score)	5 (16.67)	8 (26.67)	9 (30.00)
	improved (2 score)	0	0	2 (6.67)
*p*-value (between groups)		0.608	0.793	0.883

^a^ Numbers of patients (percentage).

**Table 5 antibiotics-10-00108-t005:** Baseline characteristics in 45 volunteers.

Baseline Characteristics	Rhodomyrtone Serum
Age (year)	25.60 ± 6.11 ^a^
Sex	
Male	11 (24.44) ^b^
Female	34 (75.56)
Investigator Global Assessment (IGA)	
1	13 (28.89)
2	32 (71.11)
3	0
4	0

^a^ Mean ± SD; ^b^ Numbers of volunteers (percentage).

**Table 6 antibiotics-10-00108-t006:** Improvement in inflammatory acne lesions in 45 volunteers.

Improvement in Inflammatory Acne	Baseline	Day 7
Inflammatory Acne Lesions		
Total	3.96 ± 4.32 ^a^	0.89 ± 2.14
Papule	2.13 ± 3.06	0.58 ± 1.78
Pustule	1.82 ± 2.45	0.31 ± 0.82
Reduction of Inflammatory Acne Lesions		
Completed		31 (68.89) ^b^
Improved		13 (28.89)
No change		1 (2.22)
**IGA scale**		
Improved		37 (82.22)
2 Score		18 (40.00)
1 Score		19 (42.22)
No change		8 (17.78)

^a^ Mean ± SD; ^b^ Numbers of volunteers (percentage).

## Data Availability

All data were included in the manuscript.

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
