# Peer review of "Rhodomyrtone as a New Natural Antibiotic Isolated from Rhodomyrtus tomentosa Leaf Extract: A Clinical Application in the Management of Acne Vulgaris"

_antibiotics, 2021, doi:10.3390/antibiotics10020108_

Round 1

Reviewer 1 Report

Review of the article: “Rhodomyrtone as a New Natural Antibiotic Isolated from Rhodomyrtus tomentosa Leaf Extract: A Clinical Application in the Management of Acne Vulgaris.”

Submission ID antibiotics-1074886

The manuscript is interesting and is in the scope of the journal. The experiments were well planned and performed. The obtained results are very promising. In my opinion the authors proposed interesting, alternative agent for treatment of acne vulgaris. The article will be interesting for a broad audience.

Abstract

Lines 21-24 – in my opinion the two first sentences of abstract should be rephrased. It must be clear for readers what was the aim of the study (presented in the sentence number 2) and what is taken from literature (first sentences). When I first read the manuscript I thought that the authors also compared activity of rhodomyrtone with vancomycin and daptomycin.

Line 25 „in each group” – in my opinion, the groups should be clearly defined. I am a bit confused. If I good understand there were two groups of 30 volunteers. One of this group used rhodomyrtone and the other one clindamycin gel. I would be grateful if it could be presented more clearly.

Introduction

Introduction is interesting, informative and generally well written. However, I have one comments, and I would be grateful if the authors could present their opinion.

In lines 43-44 the authors have written: “Natural products are rich source of antibacterial compound and majority of marketed drugs are derived from natural products”

Of course I agree with information presented in the first part of this sentence – natural products are rich source of antibacterial agents. However, I do not share optimism presented in the other part of this opinion, that majority of marketed drugs are derived from natural sources. It is truth for antibiotics that are produced by bacteria or fungi. But generally (In my opinion) most of medicines are synthetic substances. Rhodomyrtone is extracted from leaves of Rhodomyrtus tomentosa . Many plants produce metabolites that exhibit promising antimicrobial potential, they are components of e.g. essential oils. But I do not know many examples of essential oils or plant extract that are currently used in clinical practice.

Materials and methods

I think it would be important to present a chemical structure of rhodomyrtone

Lines 180-182 this part of description is not clear for me, my comments are underlined in brackets: “10 μL of rhodomyrtone (clear agent or solution in absolute ethanol?) was dissolved in the lipid suspension (do you mean lipid phase? what was the volume of this lipid suspension?) and 10 mL of Milli-Q water was warmed in water bath separately till the temperature of both phase reached 60 °C followed by addition of both and sonication for 30 min.”

Line 190 – Clinical study design – in my opinion Figure 1 should be mentioned din this part of description

Results

The results are very promising and well presented.

Discussion

The currently presented discussion is rather short. In my opinion the author should add some information about possibilities of treatment of acne vulgaris with other plant-derived substances, essential oils or extracts. Advantages of rhodomyrtone should be emphasized.

Final decision – minor revision.

Author Response

Response to Reviewer 1 Comments

Abstract

Lines 21-24 – in my opinion the two first sentences of abstract should be rephrased. It must be clear for readers what was the aim of the study (presented in the sentence number 2) and what is taken from literature (first sentences). When I first read the manuscript I thought that the authors also compared activity of rhodomyrtone with vancomycin and daptomycin.

Response 1: Corrected

Text modified. Page 1, line 21-24.

Rhodomyrtone, a plant-derived principal compound isolated from Rhodomyrtus tomentosa (Myrtaceae) leaf extract has been challenged as a potential antimicrobial drug with comparable efficacy to vancomycin and daptomycin natural alternative for the treatment of acne vulgaris.

Line 25 “in each group” – in my opinion, the groups should be clearly defined. I am a bit confused. If I good understand there were two groups of 30 volunteers. One of this group used rhodomyrtone and the other one clindamycin gel. I would be grateful if it could be presented more clearly.

Response 2: Corrected.

Text modified. Page 1, line 26-30.

In both a randomized and double-blind controlled clinical trial, 30 volunteers with mild-moderate acne severity in each group were enrolled for the study. 60 volunteers with mild-moderate acne severity were assigned into two groups: rhodomyrtone serum and clindamycin gel. and Volunteers were instructed to apply the samples to an infected area to acne lesions on their faces twice daily.

Introduction

In lines 43-44 the authors have written: “Natural products are rich source of antibacterial compound and majority of marketed drugs are derived from natural products”

Of course I agree with information presented in the first part of this sentence – natural products are rich source of antibacterial agents. However, I do not share optimism presented in the other part of this opinion, that majority of marketed drugs are derived from natural sources. It is truth for antibiotics that are produced by bacteria or fungi. But generally (In my opinion) most of medicines are synthetic substances. Rhodomyrtone is extracted from leaves of Rhodomyrtus tomentosa. Many plants produce metabolites that exhibit promising antimicrobial potential, they are components of e.g. essential oils. But I do not know many examples of essential oils or plant extract that are currently used in clinical practice.

Response 3: Corrected

Text modified. Page 2, line 48-50.

Natural products are rich source of antibacterial compounds. and majority of marketed drugs are derived from natural products [1]. Drug derived from herbal source can play a significant role in future health care system.

Materials and methods

I think it would be important to present a chemical structure of rhodomyrtone

Response 4: Added.

Text modified. Page 5, line 217-218.

The dried extract was subjected to column chromatography to elute rhodomyrtone (Figure 1).

Figure 1. Chemical structure of rhodomyrtone.

Lines 180-182 this part of description is not clear for me, my comments are underlined in brackets: “10 μL of rhodomyrtone (clear agent or solution in absolute ethanol?) was dissolved in the lipid suspension (do you mean lipid phase? what was the volume of this lipid suspension?) and 10 mL of Milli-Q water was warmed in water bath separately till the temperature of both phase reached 60 °C followed by addition of both and sonication for 30 min.”

Response 5: Corrected

Text modified. Page 5, line 225-227.

10 µL of rhodomyrtone in ethanol was dissolved in the lipid phase and 10 mL of Milli-Q water was warmed in water bath separately till the temperature of both phase reached 60 °C followed by addition of both and sonication for 30 min.

Line 190 – Clinical study design – in my opinion Figure 1 should be mentioned in this part of description

Response 6: Added.

Text modified. Page 5, line 239-241.

Sixty volunteers were randomly given treatment with liposomal encapsulated rhodomyrtone serum and clindamycin gel, for 8 weeks (Figure 2).

Discussion

The currently presented discussion is rather short. In my opinion the author should add some information about possibilities of treatment of acne vulgaris with other plant-derived substances, essential oils or extracts. Advantages of rhodomyrtone should be emphasized.

Response 7: Corrected.

Text modified. Page 3-4, line 139-170.

In addition to its anti-acne properties, rhodomyrtone presented extremely potent and broad Gram-positive antibacterial activity with its MIC and MBC at 0.39 to 0.78 µg/mL and 0.39 to 12.5 µg/mL, respectively which are comparable with last resort antibiotics in glycopeptide and lipopeptide group [1]. Both its transient binding mode to phospholipid head groups leading to distortion of lipid packing, membrane fluidization, and induction of membrane curvature and its ability to form protein-trapping membrane vesicles are unique, making it an attractive new antibiotic candidate with a novel mechanism of action [3]. Rhodomyrtone was also demonstrated to modulate the transcription of genes involved in diaminopimelate biosynthetic pathway, associated with the biogenesis of cell envelope, transporter proteins, and nucleotide metabolism [24]. The compound exhibits activity against multidrug-resistant organisms, capsule forming- and endospore-producing bacteria [25,26]. In addition, the ability to hinder bacterial adhesion to human HaCaT keratinocytes [29], inhibit biofilm producing, and killing pathogens within biofilm was demonstrated [7,28].

Propionibacterium acnes induces inflammation through Toll-like receptor 2 (TLR2) activation on keratinocytes leading to the release of pro-inflammatory cytokines such as TNF-α, IL-6, and IL-8 [29]. Following the treatment, inflamed acne lesions distinctly improved. Currently, many cytokines, chemokines, and other pro-inflammatory mediators have gained a great deal of attention by many scientists to reveal breakthroughs in the treatment of acne [30]. Evidence suggests that rhodomyrtone significantly decreased inflammatory gene expression and the expression and secretion of inflammatory proteins by modulating MAP kinase and NF-κB signaling pathways [31]. The bioactive compound could enhance the expression of pro-inflammatory molecules including IL-6, and iNOS in simulated THP-1 monocytes with heat-killed methicillin-resistant S. aureus [32]. In addition, the efficiency of a phospholipid, a natural ingredient, in liposome has been previously shown to maintain moisturizer to the skin [33] as well as acting as an anti-inflammatory agent [34]. The liposome formulation contains some fatty acids which provide skin hydration resulting in enhancing the efficacy of acne treatment in volunteers. We can affirm that liposome Liposomal encapsulated rhodomyrtone serum showed has been demonstrated better results than marketing clindamycin gel. Furthermore, volunteers' responses suggest the potential of the prototype for commercialization for the treatment of inflammatory acne vulgaris.

Reviewer 2 Report

In the abstract some information about the origin (name of the species) of the studied compound. A little conclusion (about 2 lines) should be added to complete the abstract.

In the Introduction, botanic information on Rhodomyrtus tomentosa should be added. Are there other plants that could be a source of the studied compound or is it specific in Rhodomyrtus tomentosa?

Authors should add more information to the extraction and isolation of rhodomyrtone from Rhodomyrtus tomentosa. It is not sufficient to suggest the relative previous report.

I suggest integrating "results" and "discussion" sections in order to avoid redundant information.

The conclusion should better summarise the main results discussed in the manuscript according to the aim of the research.

Author Response

Response to Reviewer 2 Comments

In the abstract some information about the origin (name of the species) of the studied compound. A little conclusion (about 2 lines) should be added to complete the abstract.

Response 1: Corrected

Text modified. Page 1, line 21-24.

Rhodomyrtone, a plant-derived principal compound isolated from Rhodomyrtus tomentosa (Myrtaceae) leaf extract has been challenged as a potential antimicrobial drug with comparable efficacy to vancomycin and daptomycin natural alternative for the treatment of acne vulgaris.

Text modified page 1, line 39-40

Rhodomyrtone serum has been demonstrated to be effective and safe for the treatment of inflammatory acne lesions.

In the Introduction, botanic information on Rhodomyrtus tomentosa should be added. Are there other plants that could be a source of the studied compound or is it specific in Rhodomyrtus tomentosa?

Response 2: Corrected

Text modified. Page 2, line 50-55.

Rhodomyrtus tomentosa (Aiton) Hassk., an evergreen shrub belonging to the family Myrtaceae, has been used traditional medicine in southeast Asian countries. Rhodomyrtone, an acylphloroglucinol compound isolated from RhodomyrtusR. tomentosa (Myrtaceae) possess excellent bio-active bioactive properties against wild range of Gram-positive bacteria [1,2], with a novel mechanism of action [3].

Authors should add more information to the extraction and isolation of rhodomyrtone from Rhodomyrtus tomentosa. It is not sufficient to suggest the relative previous report.

Response 3: Added.

Text modified. Page 5, line 213-218.

Rhodomyrtone, a pure compound from the leaves of R. tomentosa (family Myrtaceae), was isolated according to our previous report with slight modification [35,36]. Briefly, the dried leaf powder of R. tomentosa was extracted twice with 95% ethanol at room temperature for 5 days. The extract was evaporated by using a rotary evaporator (BUCHI Rotavapor R-114, Switzerland). The dried extract was subjected to column chromatography to elute rhodomyrtone (Figure 1).

I suggest integrating "results" and "discussion" sections in order to avoid redundant information.

Response 4: Results and discussion are combined.

Text modified. Page 3-4, line 103-180.

  1. Results and Discussion

Information on volunteer characteristics related to acne presented in Table 1. Thirty volunteers initially enrolled in each treatment groups: 1% liposomal encapsulated rhodomyrtone serum and 1% clindamycin gel treated group completed the whole study protocol (Figure 12). Most of the volunteers are students with an average age of 21 years, Prince of Songkla University. Initially, baseline general characteristics of all volunteers were not significantly different between the treatment groups (p > 0.05). In addition, no differences in the number of acne lesions and Investigator Global Assessment (IGA) scale among both the groups were observed (p > 0.05) (Table 2).

All subjects showed no signs of irritation or side effects for tested topical agents. Safety evaluation of all volunteers did not present a severe score of skin allergy during treatment. However, itching occurred during the early stage of treatment around the acne lesions. This event occurred at the mild score in all groups, 3.33% in 1% liposomal encapsulated rhodomyrtone serum, and 6.67% in 1% clindamycin gel (data not presented). Most of the volunteers’ responses were observed as moderately satisfied (score = 2), being 50.00% and 43.30% very satisfied (score = 3) reported as 36.70% and 40.00% in 1% liposomal encapsulated rhodomyrtone serum and 1% clindamycin gel, respectively (Table 3).

Reduction in acne lesions after treatment for 8 weeks is presented in Figure 23. All treatment agents resulted in a similar reduction in the numbers of acne lesions. Significant reduction in total numbers of acne lesions were demonstrated in all treatment groups, compared with baseline during week 2 to week 8 (Figure 23A) (p < 0.05). For non-inflammatory lesions (Figure 23B), the reduction in total acne counts was observed in all treatment groups. It was noted that volunteers with 1% liposomal encapsulated rhodomyrtone serum demonstrated improvement in inflammatory lesions concerning baseline within a few days. We have earlier reported that 1% liposomal encapsulated rhodomyrtone serum had better efficacy than 1% clindamycin, 20% azelaic acid, and 2.5% benzoyl peroxide when employed in acne treatment [10]. During week 8, total inflamed acne lesion counts in 1% liposomal encapsulated rhodomyrtone serum treated group were significantly reduced by 36.36%, comparable with 34.70% in 1% clindamycin gel (p < 0.05) (Figure 23C). A volunteer case with mild severity acne vulgaris demonstrated a remarkable reduction in the numbers of acne lesions with almost eradication of acne lesions at the end of the treatment (Figure 34). Liposomal encapsulated rhodomyrtone serum treatment in a volunteer with a moderate severity of acne vulgaris resulted in fewer numbers of acne lesions. In addition, the severity change towards mild severity at the end of treatment as represented in Figure 45. No significant differences in the IGA scale change among the two treatment groups were observed (p > 0.05) (Table 4).

In addition to its anti-acne properties, rhodomyrtone presented extremely potent and broad Gram-positive antibacterial activity with its MIC and MBC at 0.39 to 0.78 µg/mL and 0.39 to 12.5 µg/mL, respectively which are comparable with last resort antibiotics in glycopeptide and lipopeptide group [1]. Both its transient binding mode to phospholipid head groups leading to distortion of lipid packing, membrane fluidization, and induction of membrane curvature and its ability to form protein-trapping membrane vesicles are unique, making it an attractive new antibiotic candidate with a novel mechanism of action [3]. Rhodomyrtone was also demonstrated to modulate the transcription of genes involved in diaminopimelate biosynthetic pathway, associated with the biogenesis of cell envelope, transporter proteins, and nucleotide metabolism [24]. The compound exhibits activity against multidrug-resistant organisms, capsule forming- and endospore-producing bacteria [25,26]. In addition, the ability to hinder bacterial adhesion to human HaCaT keratinocytes [29], inhibit biofilm producing, and killing pathogens within biofilm was demonstrated [7,28].

Propionibacterium acnes induces inflammation through Toll-like receptor 2 (TLR2) activation on keratinocytes leading to the release of pro-inflammatory cytokines such as TNF-α, IL-6, and IL-8 [29]. Following the treatment, inflamed acne lesions distinctly improved. Currently, many cytokines, chemokines, and other pro-inflammatory mediators have gained a great deal of attention by many scientists to reveal breakthroughs in the treatment of acne [30]. Evidence suggests that rhodomyrtone significantly decreased inflammatory gene expression and the expression and secretion of inflammatory proteins by modulating MAP kinase and NF-κB signaling pathways [31]. The bioactive compound could enhance the expression of pro-inflammatory molecules including IL-6, and iNOS in simulated THP-1 monocytes with heat-killed methicillin-resistant S. aureus [32]. In addition, the efficiency of a phospholipid, a natural ingredient, in liposome has been previously shown to maintain moisturizer to the skin [33] as well as acting as an anti-inflammatory agent [34]. The liposome formulation contains some fatty acids which provide skin hydration resulting in enhancing the efficacy of acne treatment in volunteers. We can affirm that liposome Liposomal encapsulated rhodomyrtone serum showed has been demonstrated better results than marketing clindamycin gel. Furthermore, volunteers' responses suggest the potential of the prototype for commercialization for the treatment of inflammatory acne vulgaris.

Clinical trial on a prototype for commercialization was assessed in 45 subjects for a period of 1 week. Volunteer baseline characteristics related to acne was displayed in Tables 5. All volunteers presented no signs of irritation or side effects during the treatment. The clinical evaluation of the prototype demonstrated 68.89% complete reduction and 28.89% improvement in inflammatory acne lesions, with only 1 non-improved case. In most cases, the improvement in inflamed acne became obvious within 3 days following the treatment, with no severe irritation observed (Figure 56). Most subjects had IGA scale improvement, 40.00% for 2 score improvement and 42.22% for 1 score improvement (Table 6 and Figure S2). Satisfaction level assessment demonstrated 71.11% and 28.89% were at very satisfied and moderate satisfied level (Table 3).

The conclusion should better summarise the main results discussed in the manuscript according to the aim of the research.

Response 5: Corrected

Text modified. Page 7, line 311-319.

  1. 54. Conclusions

Rhodomyrtone product has been demonstrated to be effective nature derived antibiotics for treating acne lesions, especially inflammatory lesions, with comparable efficacy to clindamycin gel. Rhodomyrtone serum has been demonstrated to be effective and safe for the treatment of inflammatory acne lesions. In addition to infection control, uses in human volunteers clinically demonstrated good healing activity and skin whitening. All subjects showed no signs of irritation or long-term undesirable side effects. We believe that rhodomyrtone containing product could be used as an alternative agent for acne treatment.